

# The lncRNA MAGI2-AS3 in peripheral blood mononuclear cells: a valuable biomarker for diagnosis and prognosis prediction of breast cancer

Shanmei Du[1,*], Hong Yang[2,*], Yong Yang[3] and Kui Liu[4]

[1] School of Medical Technology, Zibo Vocational Institute, Zibo, Shandong, China
[2] Laboratory Medicine, The Affiliated Taian City Central Hospital of Qingdao University, Tai'an, Shandong Province, China
[3] Laboratory Medicine, The Second Affiliated Hospital of Shandong First Medical University, Tai'an, Shandong Province, China
[4] Center of Translational Medicine, Zibo Central Hospital, Zibo, Shandong Province, China
* These authors contributed equally to this work.

Corresponding authors
Yong Yang,
yangyong198203@163.com
Kui Liu, 11430@zbvc.edu.cn

## ABSTRACT

**Background:** Existing research has shown that long non-coding RNA (lncRNA) MAGI2 antisense RNA3 (MAGI2-AS3) expression is significantly decreased in breast cancer tissues and can inhibit breast cancer progression. However, the relationship between MAGI2-AS3 expression levels in peripheral blood mononuclear cells (PBMCs) and breast cancer remains unclear. This study aimed to explore the clinical significance of MAGI2-AS3 expression in PBMCs for diagnosing breast cancer and predicting patient prognosis.
**Methods:** Real-time quantitative reverse transcription polymerase chain reaction (qRT-PCR) was used to detect MAGI2-AS3 expression in PBMCs from healthy donors and breast cancer patients. The chi-square test analyzed the associations between MAGI2-AS3 expression and breast cancer clinicopathological parameters. The Kaplan-Meier method evaluated the impact of MAGI2-AS3 on patients' overall survival (OS), and receiver operating characteristic (ROC) analysis assessed its diagnostic accuracy for breast cancer.
**Results:** MAGI2-AS3 expression was significantly downregulated in breast cancer patients' PBMCs compared to the control group. Its expression decreased with the advancement of tumor-node-metastasis (TNM) stage and elevation of pathological grade, and was remarkably lower in patients with distant metastasis (DM). Low MAGI2-AS3 expression in PBMCs was correlated with shorter OS. ROC curve analysis showed that MAGI2-AS3 in PBMCs had good diagnostic accuracy.
**Conclusion:** MAGI2-AS3 expression in breast cancer patients' PBMCs is reduced and negatively correlated with patient outcomes. Thus, it has the potential to be a valuable biomarker for breast cancer diagnosis and prognostic evaluation.

---

## INTRODUCTION

Breast cancer has emerged as a critical global health challenge. The latest data from "Cancer statistics, 2025" in CA: A Cancer Journal for Clinicians reveal that breast cancer continues to affect a large number of people, with a significant number of new diagnoses and related deaths (*Siegel et al., 2025*). These figures firmly establish breast cancer as one of the major malignancies in humans (*Giaquinto et al., 2024*). When breast cancer is not detected early, it can progress and metastasize, worsening the prognosis. Early diagnosis and treatment are key to improving patient survival rates (*Pratt et al., 2021*). Currently, clinical breast cancer diagnosis mainly depends on pathological examination after puncture or tumor resection. However, the high cost and invasiveness of these methods make them difficult for patients to accept (*Dodelzon et al., 2024*). Moreover, nearly 30% of clinically confirmed breast cancer patients die from distant metastases (*Bartsch et al., 2024*; *Fan et al., 2025*). Thus, identifying new breast cancer tumor markers is crucial. It can aid in early breast cancer screening and the development of new treatment strategies to prevent metastasis (*Davidson, Croessmann & Park, 2021*; *Fredolini et al., 2020*).

LncRNAs represent a category of RNA molecules that lack protein-coding capabilities but assume a pivotal role in the development and progression of numerous diseases, tumors included (*Pandey & Kanduri, 2022*; *Zhang et al., 2024*). Prior research has demonstrated that abnormal expression of lncRNAs is detectable in a wide array of tumors (*Zhou et al., 2023a*). These lncRNAs can either facilitate or impede tumor progression, and their expression levels are indicative of tumor prognosis. Owing to the distinctive characteristics of these lncRNAs within cancer tissues, they are also regarded as potential biomarkers capable of predicting cancer development. In a previous investigation, our data indicated that MAGI2-AS3 exhibited reduced expression in breast cancer, and it was capable of inhibiting the progression of breast cancer (*Du et al., 2019*; *Yang et al., 2018*). These findings imply that MAGI2-AS3 holds promise as a biomarker for breast cancer diagnosis.

The detection of tumor markers in peripheral blood has long been considered an ideal approach for breast cancer screening (*Banys-Paluchowski et al., 2022*). Nevertheless, early-stage tumor cells release markers at extremely low levels in plasma or serum, resulting in low sensitivity for the detection of early-stage breast cancer. Consequently, the detection of plasma or serum is not more effective in identifying the onset of early-stage breast cancer (*Zhang et al., 2025*). During the initiation of tumorigenesis, peripheral blood nucleated cells, which are involved in the human defense mechanism, exhibit a specific stress response. Such microscopic alterations can be manifested by an increase or decrease in the expression of certain functional genes (*Li et al., 2024*). *Bitarafan et al. (2019)* reported that changes in the expression of lncRNA H19 in peripheral blood mononuclear cells (PBMCs) can serve as an important marker for predicting the occurrence of coronary artery disease. Therefore, in this study, we selected the PBMCs of breast cancer patients and analyzed the value of MAGI2-AS3 in breast cancer diagnosis and prognosis. This research is of great significance for promoting and enhancing the diagnosis of breast cancer.

## MATERIALS AND METHODS

### GEPIA

The GEPIA database (http://gepia.cancer-pku.cn/) is an online network that encompasses normal samples and tumor tissue sourced from TCGA and GTEx (*Tang et al., 2017*). We utilized this database to analyze the expression levels of MAGI2-AS3 in breast cancer tissues. The remaining default settings of the operating system were accepted.

### Kaplan-Meier plotter

The Kaplan-Meier plotter (http://kmplot.com/analysis/) was employed to evaluate how genes impact survival by analyzing cancer samples from various cancer types (*Nagy et al., 2018*). We also made use of the Kaplan-Meier plotter to analyze the relationship between MAGI2-AS3 expression and the survival of breast cancer patients. The remaining default settings of the operating system were adopted.

### Patients and sample collection

All 100 peripheral blood samples from breast cancer patients were sourced from Zibo Central Hospital. These patients were suspected of having breast cancer at initial diagnosis and had not received any treatment before blood collection. In accordance with the approval of the Institutional Review Board of Zibo Central Hospital (approval number: 202102005), written informed consent was waived for this study. As detailed in the attached "Proof of Exemption from Written Consent for Human Studies," the peripheral blood collection in this research is a common minimally invasive clinical operation with minimal risks to subjects, such as mild pain, slight bleeding at the puncture site, and a very low-probability of local infection, which are no more than the minimal risks in daily medical examinations. Also, strict privacy protection measures were implemented, including anonymizing all collected blood samples. Therefore, verbal informed consent was obtained during blood collection, and patient anonymity was ensured. Blood samples (5 ml) were collected from fasting participants in the morning using 21-gauge BD Vacutainer needles and transferred to sterile 15-ml polypropylene centrifugation tubes (CF430052; Corning, Corning NY, USA).

PBMCs were isolated from both patients and 100 age-matched healthy female volunteers (controls) *via* density gradient centrifugation with Ficoll-Hypaque (10771; Amersham Pharmacia Biotech, Uppsala, Sweden). All experiments in this study, including sample collection, RNA isolation, qRT-PCR analysis, and data processing, were conducted at the Center of Translational Medicine, Zibo Central Hospital. This facility provided the necessary equipment and environment for the research, ensuring the accuracy and reliability of the experimental results. Briefly, the process involved gently mixing an equal volume of isopycnic Ficoll-Hypaque with venous blood. The mixture was then centrifuged at 1,500 rpm for 30 min at 20–22 °C to collect the middle white membrane layer. Isolated PBMCs were quickly frozen in liquid nitrogen vapor for about 10 min and then transferred to a −80 °C freezer for storage until use (Fig. 1). The collected peripheral blood samples and isolated PBMCs were not fixed during the entire experimental process.

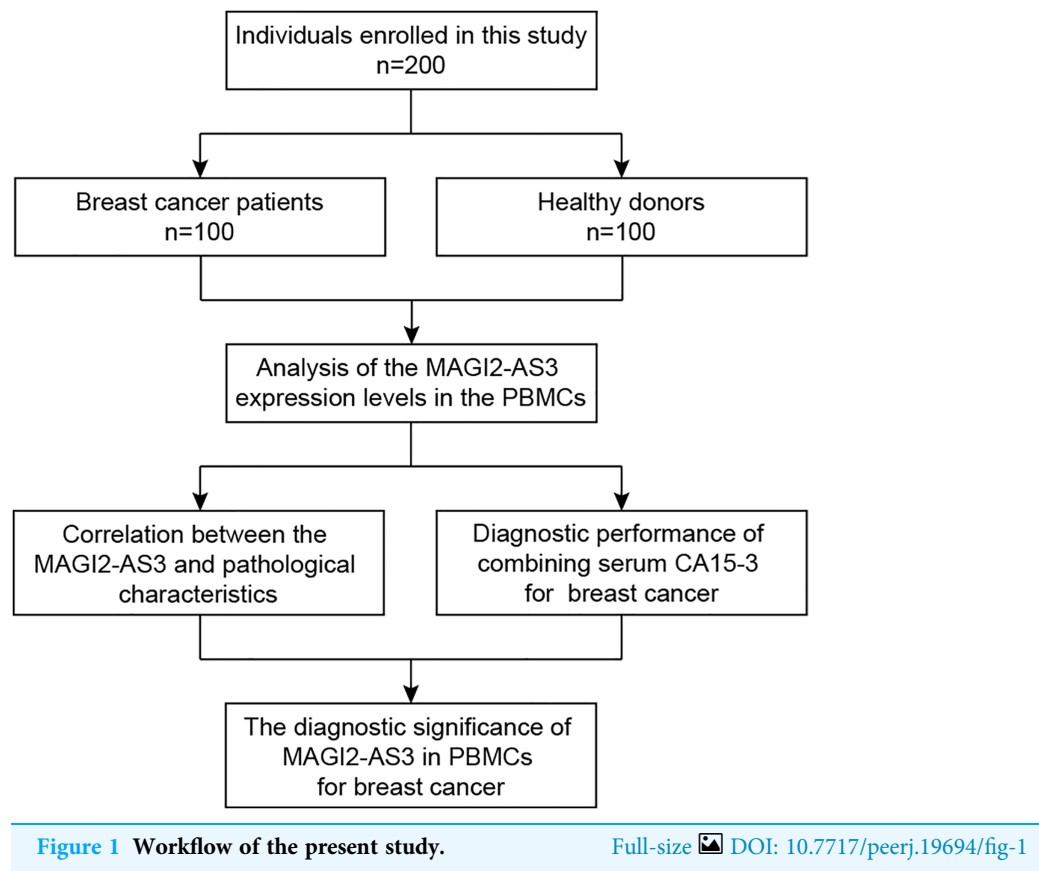

**Figure 1 Workflow of the present study.**

Patient clinical characteristics, including age, tumor size, estrogen receptor (ER) expression, progesterone receptor (PR) expression, human epidermal growth factor receptor-2 (Her-2) expression, tumor-node-metastasis (TNM) stage, and histologic grade, were retrieved from the corresponding pathology reports of the Pathology Division. The data was cross-checked for accuracy by two independent researchers. CA15-3 was detected using electrochemiluminescence methods in the clinical laboratory at Zibo Central Hospital, following the manufacturer's instructions for the Roche Cobas e601 analyzer. Quality control samples with known CA15-3 concentrations were run in parallel with patient samples to ensure the reliability of the assay. The study was approved by the Institutional Review Board of Zibo Central Hospital (approval number: 202102005) and adhered to the Declaration of Helsinki.

## RNA isolation

Total RNA from PBMCs was extracted using RNAiso Plus (9108; TAKARA Biotech, Beijing, China) following the manufacturer's protocol. Chloroform (496189; Sigma-Aldrich, Burlington, MA, USA), isopropanol (100272; Merck, Darmstadt, Germany), and 75% ethanol (prepared from absolute ethanol purchased from Sinopharm Chemical Reagent Co., Ltd., Shanghai, China) were used during the extraction process. After thawing the frozen PBMCs on ice, an appropriate volume of RNAiso Plus was added for cell lysis.

Chloroform was then added, and the mixture was centrifuged. The upper aqueous phase containing RNA was transferred, and isopropanol was used to precipitate RNA. The RNA pellet was washed with 75% ethanol, air-dried, and dissolved in RNase-free water.

RNA concentration and purity were measured using a NanoDrop 2000 spectrophotometer (Thermo Fisher Scientific, Waltham, MA, USA). The A260/A280 ratio of all samples ranged from 1.8–2.0, indicating high-quality RNA. The average RNA yield was 50 μg per $1 \times 10^6$ PBMCs, with a range of 21–75 μg per $1 \times 10^6$ PBMCs across all samples. RNA integrity was evaluated with an Agilent 2100 Bioanalyzer (Agilent Technologies, Santa Clara, CA, USA), and the RNA Integrity Number (RIN) of each sample was above 7.0. To eliminate potential genomic DNA contamination, RNA samples were treated with DNase I (EN0521; Thermo Fisher Scientific, Waltham, MA, USA) according to the manufacturer's instructions. To assess the integrity of the RNA and the effectiveness of DNA contamination removal, agarose gel electrophoresis was performed on a 1% agarose gel containing ethidium bromide (E7637; Sigma-Aldrich, Burlington, MA, USA) at 120 V for 30 min. Visual inspection of the gel during the experiment indicated that the RNA samples showed clear 28S and 18S rRNA bands, suggesting intact RNA. Additionally, no visible DNA bands were detected, which confirmed the absence of DNA contamination in the RNA samples after DNase I treatment.

## qRT-PCR

Reverse transcription was carried out using the PrimeScript RT reagent Kit with gDNA Eraser (RR047A; TAKARA Biotech, Beijing, China). The 20-μl reaction system contained 1 μg of total RNA, 4 μl of 5×PrimeScript Buffer 2, 1 μl of PrimeScript RT Enzyme Mix I, 1 μl of gDNA Eraser, and RNase-free dH$_2$O. The reaction conditions were 42 °C for 30 min for reverse transcription and 85 °C for 5 s to inactivate the enzyme. The resulting cDNA was stored at −20 °C.

qRT-PCR was performed using SYBR Premix Ex Taq II (RR820A; TAKARA Biotech, Beijing, China) on an ABI 7500 Real-Time PCR System (Thermo Fisher Scientific, Waltham, MA, USA). The 20-μl reaction system included 10 μl of SYBR Premix Ex Taq II, 0.8 μl of each forward and reverse primer (10 μM), 2 μl of cDNA template, and 6.4 μl of ddH$_2$O. All of the primers used are listed in Table 1.

The thermocycling parameters were 95 °C for 30 s of initial denaturation, followed by 40 cycles of 95 °C for 5 s and 60 °C for 30 s. Melting curve analysis was conducted at the end to verify product specificity, showing a single peak. A no-template control (*Cameron et al., 2017*) was included in each run, and no amplification was detected in NTC wells.

In our relative quantification study, although the main goal was to analyze the relative expression levels of MAGI2-AS3 and β-actin, we still generated standard curves using serial dilutions ($10^8$–$10^3$ copies/μl) of plasmids containing the target gene sequences. These standard curves were essential for evaluating the performance of our qRT-PCR assays. For MAGI2-AS3, the standard curve had a slope of −3.32, a y-intercept of 37.25, an $R^2$ value of 0.996, and a PCR efficiency of 98.6%. For β-actin, the slope was −3.30, the y-intercept was 37.40, the $R^2$ value was 0.997, and the PCR efficiency was 99.2%. The high $R^2$ values demonstrated a reliable linear relationship between the template concentration and the

**Table 1 Sequence of primers used for qRT-PCR.**

| Gene name | Primer sequence |
|---|---|
| MAGI2-AS3 | F: 5′-CACCTTGCTTGACACACTTGA-3′ |
| | R: 5′-CATTACAGCTCGGCTACTGC-3′ |
| β-actin | F: 5′-CCTTCCTGGGCATGGAGTC-3′ |
| | R: 5′-TGATCTTCATTGTGCTGGGTG-3′ |

Ct values, validating the reproducibility of our experimental procedures. The PCR efficiencies within an appropriate range further attested to the effectiveness of our reaction conditions.

We determined the linear dynamic range of our qRT-PCR assays to be $10^3$–$10^8$ copies/μl. This range was significant as it defined the concentration interval within which we could accurately detect relative changes in gene expression. Samples with target gene expression levels falling within this range could be effectively analyzed for relative quantification.

The limit of detection (*Perozo et al., 2002*) for both MAGI2-AS3 and β-actin was found to be $10^3$ copies/μl. This LOD represented the lowest concentration at which we could confidently detect the target genes. In the context of relative quantification, this information was valuable for ensuring that the detected gene expression changes were reliable and not influenced by background noise.

Although inhibition testing is an important aspect of qRT-PCR quality control, the primary objective of this study was to rapidly explore the potential value of MAGI2-AS3 in the diagnosis and prognosis prediction of breast cancer. Our focus was on analyzing the expression differences of MAGI2-AS3 in peripheral blood mononuclear cells between breast cancer patients and healthy controls, as well as its associations with clinicopathological parameters and patient survival outcomes. Given the research priorities and resource allocation, we prioritized our efforts on the key experiments and data analysis. Although inhibition testing is valuable, it was not essential for achieving the core goals of this study, and its results would have relatively minor direct impacts on the research conclusions. Therefore, we did not perform inhibition testing in this study.

## Statistical analysis

Statistical analysis was performed using GraphPad Prism 5.0 (GraphPad Software, San Diego, CA, USA) and SPSS version 17 (IBM, Armonk, New York, USA). Student's t-test was used for comparing two groups, and one-way ANOVA was applied for multiple-group comparisons. The chi-square test was used to analyze the correlation between MAGI2-AS3 expression and clinicopathological variables. The log-rank test was used for comparing patient survival curves, and the *receiver operating characteristic (ROC)* curve was used to evaluate the diagnostic value of MAGI2-AS3. A *p*-value < 0.05 was considered statistically significant. Outliers were identified using Grubbs' test. Data were normalized using the $2^{-\Delta\Delta Ct}$ method with β-actin as the reference gene. Each sample was analyzed in three biological replicates and three technical replicates. The coefficient of variation (CV) was calculated to assess repeatability (intra-assay variation) and reproducibility (inter-assay variation). The intra-assay CV for MAGI2-AS3 expression was less than 5%, and the

inter-assay CV was less than 10%. A power analysis was conducted using G*Power software. Based on an expected effect size, a significance level of $\alpha = 0.05$, and a power of $1-\beta = 0.8$, the sample size of 100 in each group was determined to be sufficient for detecting significant differences.

# RESULTS

## MAGI2-AS3 is downregulated in breast cancer

Using the GEPIA database, we found that MAGI2-AS3 expression was significantly downregulated in breast cancer tissues compared to normal tissues (Figs. 2A and 2B). Additionally, we analyzed the prognostic potential of MAGI2-AS3 using the Kaplan-Meier plotter database. Reduced MAGI2-AS3 expression was significantly associated with unfavorable overall survival (OS) outcomes in breast cancer patients (Fig. 2C).

## The lncRNA MAGI2-AS3 was downregulated in the PBMCs of breast cancer patients, and its low expression was associated with a shorter overall survival

The expression levels of MAGI2-AS3 in the PBMCs of breast cancer patients and healthy females were measured. The RT-qPCR results demonstrated that breast cancer patients had significantly lower MAGI2-AS3 levels (Fig. 3A). Distant metastasis (DM) represents a crucial and often ominous stage in the progression of breast cancer. It refers to the spread of cancer cells from the primary tumor in the breast to distant parts of the body, most commonly the lungs, liver, bones, or brain. This metastatic spread significantly complicates treatment and reduces patients' survival prospects. In the context of our study, we specifically focused on the differences in MAGI2-AS3 expression levels between patients with and without DM. This choice was driven by the need to explore the potential of MAGI2-AS3 as a biomarker for predicting breast cancer metastasis. By analyzing these differences, we aimed to gain a deeper understanding of the molecular mechanisms involved in the metastatic process. Such insights could be instrumental in the early detection of metastasis and the development of more targeted and effective treatment strategies, ultimately improving patient outcomes. After defining the importance of DM and our research goal, we delved into the analysis and obtained the following findings. As expected, we found that MAGI2-AS3 was expressed at lower levels in the PBMCs of breast cancer patients with DM compared to those without DM (Fig. 3B). Based on the expression level of MAGI2-AS3 in PBMCs, breast cancer patients were classified into two groups. Kaplan-Meier analysis revealed that low MAGI2-AS3 expression was significantly associated with shorter OS in breast cancer patients, indicating its potential prognostic value (Fig. 3C).

## Associations between the MAGI2-AS3 expression level in PBMCs and clinicopathologic variables of breast cancer patients

The associations between MAGI2-AS3 expression and the clinical characteristics of breast cancer patients were explored using the chi-square test. MAGI2-AS3 expression was strongly correlated with tumor size, TNM stage, and histologic grade. There was no

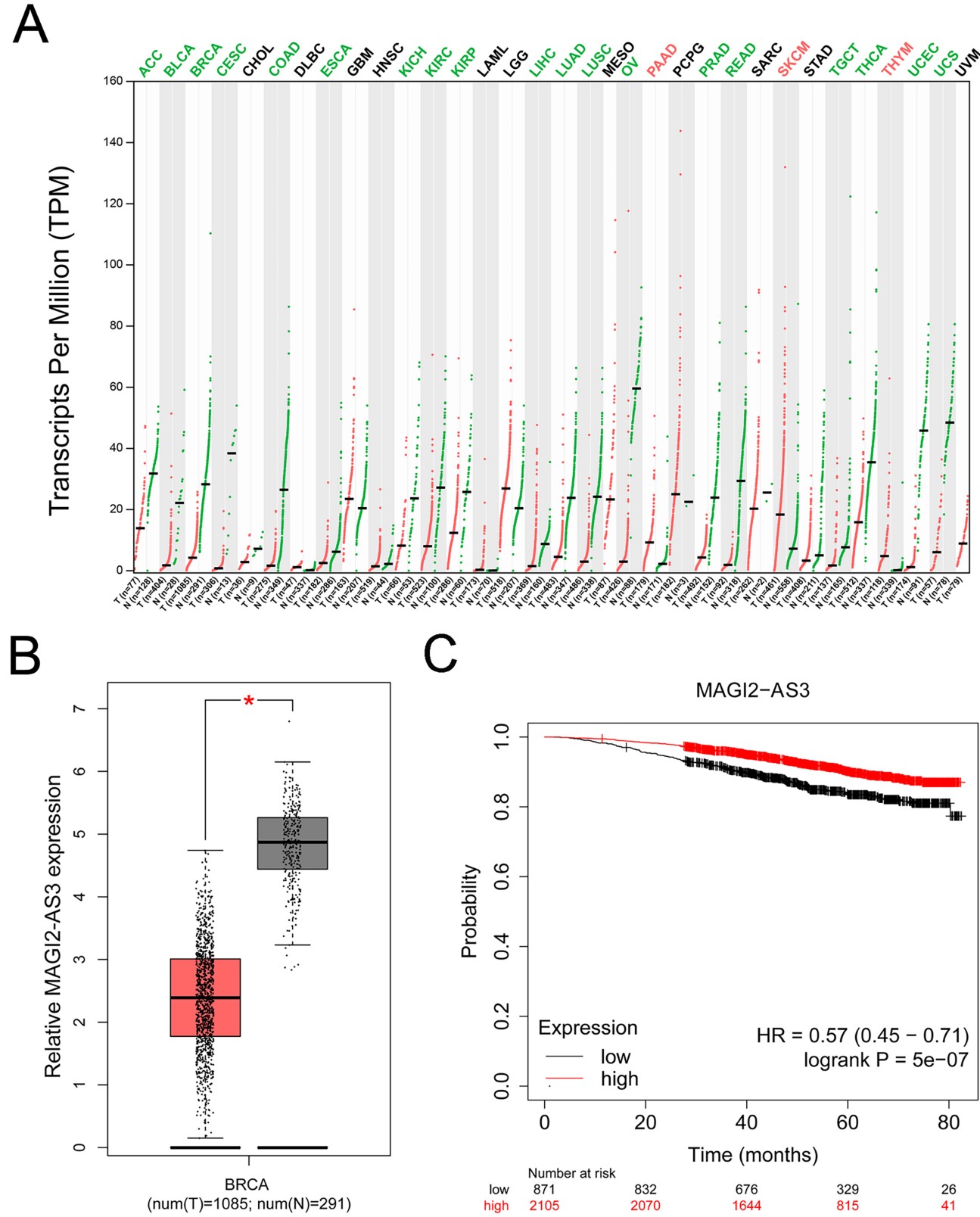

**Figure 2** **MAGI2-AS3 expression patterns and their association with breast cancer prognosis.** (A) Bioinformatics analysis using the GEPIA database. The colors of tumor names (*e.g.*, ACC, BLCA, BRCA) indicate MAGI2-AS3 expression patterns: green means MAGI2-AS3 is downregulated

# PeerJ

**Figure 2 (continued)**
in tumor tissues compared to normal tissues; red means it is upregulated; black means no significant difference. Red and green dots represent tumor and normal tissues respectively, highlighting the differential expression of MAGI2-AS3. (B) Employing GEPIA, analysis of TCGA and GTEx datasets shows that MAGI2-AS3 is downregulated in breast cancer (BRCA) tumour tissues compared to normal tissues (*$p < 0.05$). (C) Kaplan-Meier analysis of breast cancer patients reveals that lower MAGI2-AS3 expression is associated with a poor overall survival prognosis (HR = 0.57 (0.45–0.71), logrank $p$ = 5e−07).

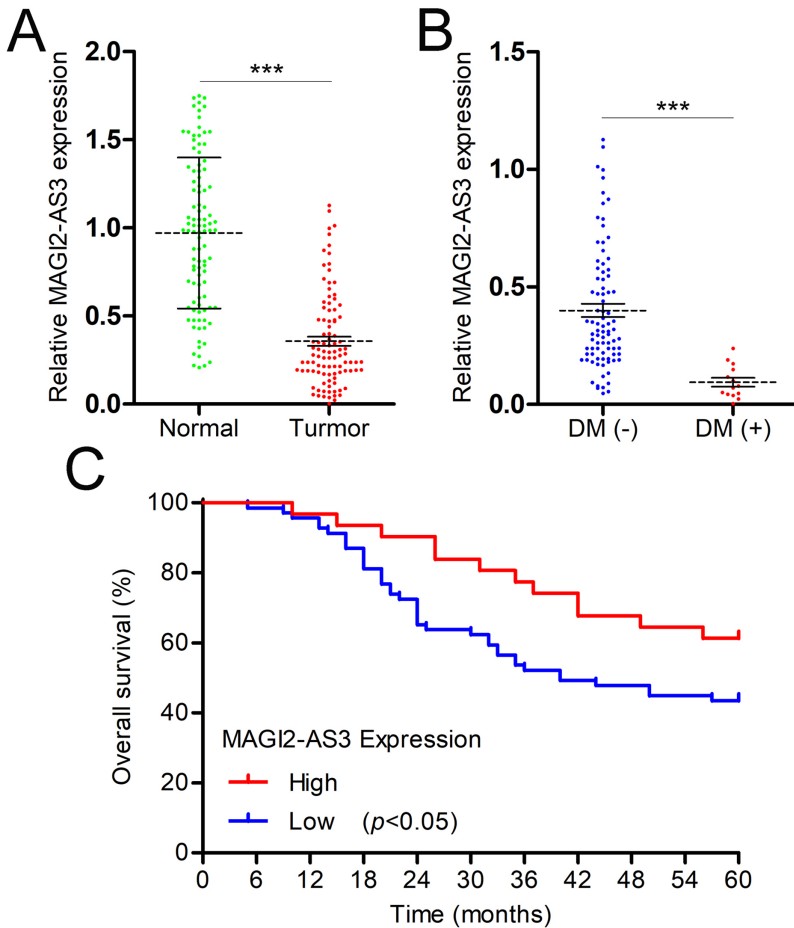

**Figure 3 MAGI2-AS3 is downregulated in the PBMCs of breast cancer patients.** (A) qRT-PCR was utilised to measure MAGI2-AS3 expression in the PBMCs of healthy females and patients with breast cancer. (B) MAGI2-AS3 expression levels among breast cancer patients with or without DM. (C) Kaplan-Meier survival curves for patients with low MAGI2-AS3 expression levels and those with high MAGI2-AS3 expression levels in breast cancer cases (***$p < 0.001$).

significant correlation between MAGI2-AS3 expression and other clinical indices, such as patient age, ER status, PR status, and HER-2 status (Table 2). The strong correlations between MAGI2-AS3 expression and tumor size, TNM stage, and histologic grade hold significant clinical implications. Tumor size is an important factor reflecting the progression of breast cancer. The significant correlation with MAGI2-AS3 expression indicates that this lncRNA may serve as a supplementary marker for evaluating tumor burden. For instance, patients with larger tumors often show lower MAGI2-AS3

**Table 2 Association of lncRNA MAGI2-AS3 with clinicopathological features of breast cancer patients.**

| Parameters | Number | lncRNA MAGI2-AS3 expression | | | |
|---|---|---|---|---|---|
| | | Low ($n$ = 69) | High ($n$ = 31) | $\chi 2$ | $p$ |
| Age, years | | | | | |
| ≤40 | 38 | 23 | 15 | 2.057 | 0.151 |
| >40 | 62 | 46 | 16 | | |
| Tumor size (cm) | | | | | |
| ≤3 | 47 | 26 | 21 | 7.760 | <0.01** |
| >3 | 53 | 43 | 10 | | |
| ER | | | | | |
| Negative | 61 | 39 | 22 | 1.876 | 0.170 |
| Positive | 39 | 30 | 9 | | |
| PR | | | | | |
| Negative | 66 | 47 | 19 | 0.444 | 0.505 |
| Positive | 34 | 22 | 12 | | |
| HER-2 | | | | | |
| Negative | 58 | 41 | 17 | 0.184 | 0.667 |
| Positive | 42 | 28 | 14 | | |
| TNM stage | | | | | |
| I–II | 55 | 31 | 24 | 9.124 | <0.01** |
| III–IV | 45 | 38 | 7 | | |
| Histologic grade | | | | | |
| I–II | 45 | 24 | 21 | 8.388 | <0.01** |
| III | 55 | 45 | 10 | | |

**Note:**
** $p$ < 0.01.

expression levels, which might be associated with more advanced disease and a worse prognosis. This finding suggests that MAGI2-AS3 could potentially assist clinicians in better assessing the severity of breast cancer at an early stage. TNM stage plays a crucial role in determining the appropriate treatment plan for breast cancer patients. The correlation between MAGI2-AS3 expression and TNM stage implies that it could be integrated into the current staging system. This integration might help in more accurately stratifying patients, especially those at the boundaries of different stages. For example, for patients with early-stage breast cancer but low MAGI2-AS3 expression, more aggressive treatment strategies could be considered to prevent disease recurrence and metastasis. Histologic grade is an indicator of the malignancy of cancer cells. The relationship between MAGI2-AS3 expression and histologic grade provides valuable insights into the biological behavior of breast cancer. Lower MAGI2-AS3 levels in tumors with higher histologic grades may be related to increased cell proliferation, invasion, and metastatic potential. This information can guide pathologists in making more precise prognostic judgments and contribute to further research on the molecular mechanisms of breast cancer progression. Although no significant correlations were found between MAGI2-AS3 expression and

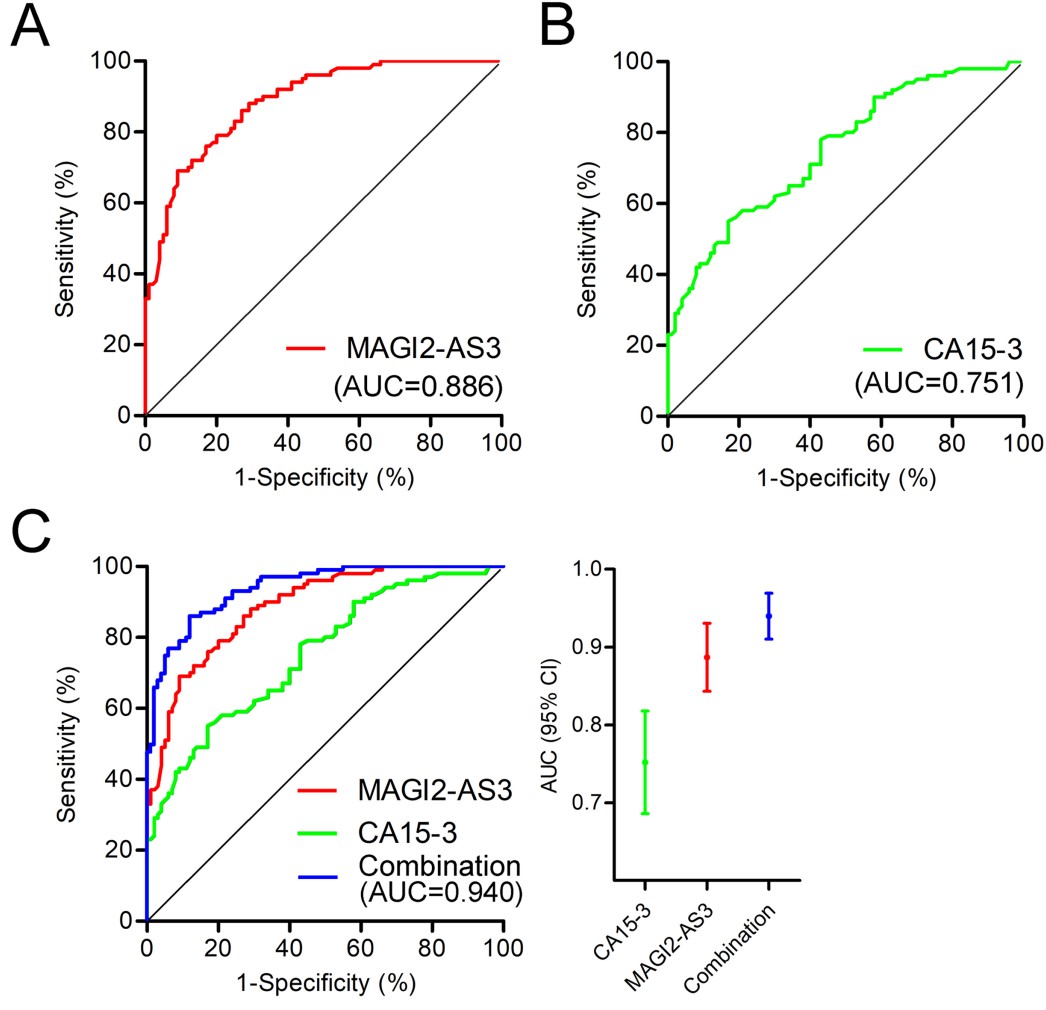

**Figure 4  Diagnostic value of MAGI2-AS3 in PBMCs for breast cancer.** (A) ROC curve of MAGI2-AS3 in PBMCs for diagnosing breast cancer, with an area under the curve (AUC) of 0.886. (B) ROC curve of serum CA15-3 for diagnosing breast cancer, with an AUC of 0.751. (C) Combined ROC curves of MAGI2-AS3 in PBMCs (red), CA15-3 in serum (green), and their combination (blue). The combination model had an AUC of 0.940, demonstrating superior diagnostic performance compared to single markers.

other clinical indices such as patient age, ER status, PR status, and HER-2 status, it is possible that there are underlying associations in specific patient subgroups. Future research with larger sample sizes and more in-depth subgroup analyses is needed to fully explore these potential relationships. This could lead to a more comprehensive understanding of the role of MAGI2-AS3 in breast cancer and potentially uncover new therapeutic targets or prognostic markers.

## Diagnostic performance of MAGI2-AS3 expression in PBMCs from breast cancer patients

In a further analysis of the diagnostic value of MAGI2-AS3 in PBMCs, we constructed a ROC curve to investigate the diagnostic value of MAGI2-AS3 in the PBMCs of breast

cancer patients. The ROC curve indicated that when comparing patients with controls, MAGI2-AS3 had an area under the curve (AUC) of 0.886 (Fig. 4A). We also constructed an ROC curve using CA15-3 data. Compared to the control group, CA15-3 had an AUC value of 0.751 (Fig. 4B). The analysis of the ROC curve showed that the diagnostic efficiency of MAGI2-AS3 was higher than that of CA15-3. These results suggested that MAGI2-AS3 has the potential to serve as a diagnostic marker for breast cancer. Moreover, the combination of MAGI2-AS3 in PBMCs and serum CA15-3 exhibited better diagnostic performance for breast cancer (AUC = 0.940) than a single marker (Fig. 4C).

## DISCUSSION

Breast cancer has become a leading global malignancy, accounting for nearly 12% of all new cancer diagnoses according to the World Health Organization (*Wilkinson & Gathani, 2022*). Early detection is vital for better treatment outcomes and reduced mortality (*Bardia et al., 2024*). Current detection methods like mammography, ultrasonography, and MRI have limitations, especially in early-stage and high-density breast tissue cases (*Eisemann et al., 2025*). Serum-based tumor markers such as CA15-3 also suffer from low sensitivity and specificity, highlighting the need for novel diagnostic approaches (*Lian et al., 2019*).

In the field of cancer research, lncRNAs have emerged as crucial players in tumorigenesis and cancer progression (*Wang et al., 2025*; *Zhou et al. 2023b*). They are involved in various biological processes and have shown great potential as diagnostic and prognostic biomarkers. Our research focuses on one such lncRNA, MAGI2-AS3. Previous studies have clearly demonstrated that MAGI2-AS3 is downregulated in breast cancer tissues (*Yang et al., 2018*). In these tissues, it exerts its inhibitory effects on the proliferation and migration of breast cancer cells through the Fas/FasL signalling pathway and the miR-374a/PTEN axis (*Du et al., 2019*). Survival analysis has also indicated that MAGI2-AS3 serves as an independent prognostic factor for breast cancer. Patients with low expression levels of this lncRNA are associated with a poorer OS. Our ultimate goal is to comprehensively understand the role of MAGI2-AS3 in breast cancer for more accurate diagnosis and prognosis assessment. Considering the low sensitivity of plasma/serum lncRNA detection in early-stage tumours, we decided to extend our research to MAGI2-AS3 in PBMCs. The results show that, compared with healthy controls, MAGI2-AS3 in PBMCs of breast cancer patients is significantly downregulated, which is consistent with its expression pattern in cancer tissues (*Xue et al., 2021*; *Yang et al., 2018*).

Moreover, further analysis reveals that MAGI2-AS3 expression in PBMCs is closely correlated with tumor burden. This includes aspects such as tumor size, TNM stage, and histologic grade. In addition, we found that MAGI2-AS3 has no significant correlation with non-cancer factors like patient age, ER, PR, or HER-2 status (as presented in Table 1). These findings, taken together, provide a more comprehensive understanding of the relationship between MAGI2-AS3 and breast cancer. Notably, the extremely low expression of MAGI2-AS3 in patients with distant metastases further highlights its potential value in predicting prognosis. The ROC analysis demonstrated that MAGI2-AS3

in PBMCs has an AUC of 0.886. When combined with serum CA15-3, the AUC increased to 0.940, significantly enhancing the diagnostic efficacy.

In addition to its prognostic significance, the potential of MAGI2-AS3 as a diagnostic biomarker for breast cancer subtypes warrants further investigation. While our current study revealed significant associations between MAGI2-AS3 expression and clinicopathological features such as tumor size, TNM stage, and histological grade, we acknowledge that the limited sample size constrained our ability to comprehensively evaluate its diagnostic accuracy across all breast cancer subtypes. This is an important consideration, as breast cancer is a heterogeneous disease with distinct molecular and pathological subtypes that may exhibit differential expression patterns of MAGI2-AS3. Future research should focus on expanding the sample size to include a broader spectrum of breast cancer subtypes, thereby enabling a more robust analysis of MAGI2-AS3's diagnostic utility. Additionally, integrating MAGI2-AS3 with other established biomarkers or clinical parameters may enhance its diagnostic performance and provide a more comprehensive approach to breast cancer diagnosis and prognosis.

The underlying mechanisms of MAGI2-AS3 in PBMCs are yet to be fully understood. It may be involved in cell-cell communication, with tumor-derived exosomes potentially "educating" PBMCs and downregulating MAGI2-AS3, thus affecting immune-regulatory functions (*Akoto & Saini, 2021*). In the immune system, MAGI2-AS3 could modulate genes related to immune cell activation and cytokine production, such as IL-6 and TNF-$\alpha$, which are crucial in tumor-associated inflammation. Epigenetic modifications might also play a role, with MAGI2-AS3 potentially recruiting modifiers to regulate genes related to breast cancer development (*Kai-Xin et al., 2021*; *Shaath et al., 2022*). In summary, this study establishes MAGI2-AS3 in PBMCs as a promising non-invasive biomarker for breast cancer, supported by its distinct expression patterns, clinical correlations, and diagnostic performance. While the underlying mechanisms linking MAGI2-AS3 to tumor progression and immune regulation require further clarification, the current findings provide a critical foundation for developing non-invasive screening and prognostic tools. To fully validate its clinical utility, future research should prioritize multi-center studies with larger sample sizes to assess its performance across diverse breast cancer subtypes. Additionally, mechanistic investigations into how MAGI2-AS3 modulates immune cell function and tumor-immune interactions will be essential to translate these findings into targeted therapeutic strategies, ultimately advancing precision medicine in breast cancer management.

## CONCLUSIONS

In conclusion, MAGI2-AS3 in PBMCs holds great promise as a biomarker for breast cancer diagnosis and prognosis. However, due to the limited sample size, further large-scale studies and long-term follow-up are needed to validate its clinical value. Unraveling its mechanisms could also open doors for targeted therapies.

## Funding

This work was supported by grants from the Shandong Province Natural Science Foundation of China (No. ZR2021MH024, No. ZR2023MC181), Shandong Province Key Research and Development Plan (No. 2019GSF108195), Zibo Municipal Medical and Health Technology Project (Project Number: 20240309032), Tai'an Science and Technology Development Planning Project (No. 2018NS0220) and the National Natural Science Foundation of China (No. 81602330). The funders had no role in study design, data collection and analysis, decision to publish, or preparation of the manuscript.

## Grant Disclosures

The following grant information was disclosed by the authors:
Shandong Province Natural Science Foundation of China: ZR2021MH024, ZR2023MC181.
Shandong Province Key Research and Development Plan: 2019GSF108195.
Zibo Municipal Medical and Health Technology Project: 20240309032.
Tai'an Science and Technology Development Planning Project: 2018NS0220.
National Natural Science Foundation of China: 81602330.

## Competing Interests

The authors declare that they have no competing interests.

## Author Contributions

- Shanmei Du performed the experiments, analyzed the data, prepared figures and/or tables, and approved the final draft.
- Hong Yang performed the experiments, analyzed the data, prepared figures and/or tables, and approved the final draft.
- Yong Yang conceived and designed the experiments, analyzed the data, authored or reviewed drafts of the article, and approved the final draft.
- Kui Liu conceived and designed the experiments, analyzed the data, authored or reviewed drafts of the article, and approved the final draft.

## Human Ethics

The following information was supplied relating to ethical approvals (*i.e.*, approving body and any reference numbers):

The study was approved for use by the Institutional Review Board of Zibo Central Hospital (Zibo, China; approval number: 202102005).

## Data Availability

The raw measurements are available in the Supplemental File.

## Supplemental Information

Supplemental information for this article can be found online at http://dx.doi.org/10.7717/peerj.19694#supplemental-information.

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
