# Peer review of "The lncRNA MAGI2-AS3 in peripheral blood mononuclear cells: a valuable biomarker for diagnosis and prognosis prediction of breast cancer"

_PeerJ, doi:10.7717/peerj.19694_

## Round 0.1 · original submission · Major Revisions

· Academic Editor

Major Revisions

**Language Note:** The review process has identified that the English language must be improved. PeerJ can provide language editing services - please contact us at [email protected] for pricing (be sure to provide your manuscript number and title). Alternatively, you should make your own arrangements to improve the language quality and provide details in your response letter. – PeerJ Staff

Reviewer 1 ·

Basic reporting

The manuscript is well written in professional English. Though there is a scope of improvement in a few instances.

Comments:
1. Line 137: "The data was cross-checked..."
2. Line 242: The phrase "PBMCs of breast cancer patients" is repeated twice.
3. Line 250: The sentence is incomplete: "Kaplan-Meier analysis revealed that low MAGI2-AS3..."

Experimental design

Though the investigation performed is not that rigorous, the research question is well defined. And the analyses performed is sufficient to answer the question. Just one concern:

Why was MAGI2-AS3 alone considered for this study? Are there any other lncRNA with a promise of being a good biomarker for breast cancer? Can MAGI2-AS3 diagnose all sub types of breast cancer?

Validity of the findings

The conclusion section can be expanded a bit more. Association of MAGI2-AS3 with clinical features can be included.

Additional comments

In line 118 it is mentioned that verbal informed consent was obtained. The authors have included a proof of not requiring written consent. This should be written in the text referring to the document attached.

In line 248 with DM and without DM is mentioned. DM (distant metastasis) is first mentioned in the Abstract. It is nowhere mentioned in the text before with an explanation what it means. Explain about it in the text, and why it was chosen.

Reviewer 2 ·

Basic reporting

The authors should check carefully about the gramma and figure captions for the manuscript.
Major issues:
1. Figure 2A figure caption is unclear, what are the differences of the green, red and black cancer types? It’s too redundant and not to the point. What is the meaning of red and green dots in the plot?
2. Figure 2B, what is the test being used and the test statistics and p-value?
3. Figure 3 is too blur and not able to see any text within the figures.
4. Line 150, Section 3.2 is not finished.
5. Figure 5, C/D, the caption was missing.
Minor gramma issues:
1. 78-79, “LncRNAs represent a category of RNA molecules that lack protein-coding capabilities and assume a pivotal role…”, the authors should consider change “and” to “but” for smooth transition.
2. 227, “The findings indicated that MAGI2-AS3 was remarkably down regulated in the majority of tumor tissue types, breast cancer tissues included”, the correct gramma is “including breast cancer tissues”.
3. Figure 1, the box in the third row of the workflow. “Analysis the MAGI2-AS3…” should be “Analysis of the MAGI2-AS3...”.

Experimental design

The research question is well defined. The experimental design was well described.

Validity of the findings

The authors should remove some redundant analysis that are not related to the conclusion and be more accurate and detailed about their findings.
1. In line 237-239, what are those genes function and how are they related to breast cancer? Why the authors said a negative correlation was observed between all the genes to MAGI2-AS3? It is clear to me that some genes have positive R. The authors should consider correct their conclusions. Further, the association between MAGI2-AS3 and other marker genes does not necessarily prove that MAGI2-AS3 can also be a marker gene, the authors should discuss that in discussion or consider removing this analysis.
2. Figure 5B is really not necessary to the conclusion, it only shows CA15-3 is a biomarker, which has already been reported in other manuscripts, the authors should consider remove that figure and only focus on comparison of MAGI2-AS3 to CA15-3.

Additional comments

Breast cancer remains a major global health challenge, with early detection crucial for improving patient outcomes. Current diagnostic methods (e.g., mammography, serum markers like CA15-3) have limitations, including invasiveness and low sensitivity for early-stage tumors. Long non-coding RNAs (lncRNAs) are emerging as key regulators in cancer. MAGI2-AS3, a lncRNA previously shown to be downregulated in breast cancer tissues and to inhibit tumor progression, has not been thoroughly studied in peripheral blood mononuclear cells (PBMCs). This gap motivates exploring its potential as a non-invasive biomarker.

The study aims to address the need for reliable, non-invasive biomarkers to improve early diagnosis and prognosis prediction in breast cancer. By focusing on MAGI2-AS3 in PBMCs, the authors propose a novel approach to overcome the limitations of tissue-based diagnostics and existing serum markers.

By bioinformatics analysis of PBMCs from 100 breast cancer patients and 100 healthy controls. The authors found:
1. MAGI2-AS3 expression in PBMCs was significantly lower in breast cancer patients compared to controls and correlated with advanced TNM stage, larger tumor size, and poorer histologic grade.
2. Low MAGI2-AS3 levels were associated with shorter overall survival.
3. Diagnostic accuracy of MAGI2-AS3 alone (AUC = 0.886) surpassed CA15-3 (AUC = 0.751), and their combination achieved superior performance (AUC = 0.940).
4. The study highlights MAGI2-AS3 in PBMCs as a promising non-invasive biomarker for breast cancer diagnosis and prognosis. However, larger cohorts and mechanistic studies are needed to validate clinical utility and explore underlying pathways.

While the research topic of this manuscript is interesting and the research is fine designed. I have concerns about the manuscript writing and that the authors are not showing too much unrelated results for the conclusion.

---

## Round 0.2 · accepted · Accept

· Academic Editor

Accept

All issues pointed by the reviewers were adequately addressed and the revised manuscript is acceptable now.

Reviewer 1 ·

Basic reporting

After the revisions, the article is improved and can be sent for production.

Experimental design

Though there was no rigorous investigation, the performed analyses are sufficient to address the issue.

Validity of the findings

All underlying data has been provided.

Reviewer 2 ·

Basic reporting

The language has been improved a lot since last version. It is now clear and unambiguous.

Experimental design

The authors have addressed all my concerns on the experimental design. Now the design is appropriate and rigorous.

Validity of the findings

The authors have addressed all my concerns on the results section. Now the results are robust and valid.

Additional comments

The authors have addressed all my comments. The manuscript is now good to publish.